# Resolving the Mechanism of Acoustic Plasmon Instability in Graphene Doped by Alkali Metals

**DOI:** 10.3390/ijms23094770

**Published:** 2022-04-26

**Authors:** Leonardo Marušić, Ana Kalinić, Ivan Radović, Josip Jakovac, Zoran L. Mišković, Vito Despoja

**Affiliations:** 1Maritime Department, University of Zadar, M. Pavlinovića 1, 23000 Zadar, Croatia; 2Department of Atomic Physics, “VINČA” Institute of Nuclear Sciences—National Institute of the Republic of Serbia, University of Belgrade, P.O. Box 522, 11001 Belgrade, Serbia; ana.kalinic@vin.bg.ac.rs (A.K.); iradovic@vin.bg.ac.rs (I.R.); 3Institut za Fiziku, Bijenička 46, 10000 Zagreb, Croatia; jjakovac@ifs.hr (J.J.); vito@phy.hr (V.D.); 4Department of Applied Mathematics and Waterloo Institute for Nanotechnology, University of Waterloo, Waterloo, ON N2L 3G1, Canada; zmiskovi@uwaterloo.ca; 5Donostia International Physics Center (DIPC), P. Manuel de Lardizabal 4, 20018 San Sebastián, Spain

**Keywords:** plasmon, acoustic plasmon, graphene, graphene intercalation compounds, EELS

## Abstract

Graphene doped by alkali atoms (AC${}_{x}$) supports two heavily populated bands (π and σ) crossing the Fermi level, which enables the formation of two intense two-dimensional plasmons: the Dirac plasmon (DP) and the acoustic plasmon (AP). Although the mechanism of the formation of these plasmons in electrostatically biased graphene or at noble metal surfaces is well known, the mechanism of their formation in alkali-doped graphenes is still not completely understood. We shall demonstrate that two isoelectronic systems, KC${}_{8}$ and CsC${}_{8}$, support substantially different plasmonic spectra: the KC${}_{8}$ supports a sharp DP and a well-defined AP, while the CsC${}_{8}$ supports a broad DP and does not support an AP at all. We shall demonstrate that the AP in an AC${}_{x}$ is not, as previously believed, just a consequence of the interplay of the π and σ intraband transitions, but a very subtle interplay between these transitions and the background screening, caused by the out-of-plane interband C(π)→A(σ) transitions.

## 1. Introduction

Studying acoustic plasmons (APs) in single-layer [1,2,3,4,5], double-layer [3,6,7,8,9] and multi-layer [10,11,12,13] graphene or in metal/dielectric/graphene superstructures [14] is a very active field of research. Recently, an AP was detected in a graphene–dielectric–metal structure using near-field scattering microscopy [2]. That experiment demonstrated a very robust character of the AP with small propagation loss, which enables its efficient coupling to the electromagnetic field at infra-red frequencies. The graphene/graphene-nanoribbon superstructure also supports a strong AP which can be excited by light and provide strong field enhancement inside the nano-gap, allowing efficient biosensing [6]. In this work, we focus on the AP in graphene doped by alkali metals.

As alkali atoms are added to graphene deposited on various substrates, they intercalate between the graphene layer and the substrate and metalize, i.e., they arrange in a two-dimensional (2D) crystal lattice, which supports a partially filled parabolic band [15,16,17,18,19,20,21]. The alkali atoms then usually donate electrons to the graphene π band, lifting the Fermi level above the Dirac point and transforming the graphene from a gapless semiconductor into a metal. Therefore, the alkali adlayer results in the formation of two thin, i.e., quasi two-dimensional (q2D), plasmas able to support a variety of different plasmonic modes, which do not exist in pristine or electrostatically doped graphene. Our previous theoretical research of electronic excitations in graphene doped by alkali atoms (AC${}_{x}$) showed that this system supports two kinds of intraband or 2D plasmons: the Dirac plasmon (DP) and the acoustic plasmon (AP). The intensity of these modes depends on the type of the dielectric substrate, as well as on electrostatic or natural doping [22,23,24,25]. Moreover, these theoretical studies suggest the possibility of simple manipulation of the DP and AP intensities (they can be switched ‘on’ and ‘off’ in a controlled way) [25], thereby opening new possibilities for their application in various fields, such as plasmonics, photonics, transformation optics, optoelectronics, light emitters, detectors and photovoltaic devices [26,27,28,29,30,31,32,33,34,35,36]. Additionally, these ‘tunable’ 2D plasmons could be very useful in the area of chemical or biological sensing [27,37,38,39,40].

In this paper, we focus on the AP, firstly because of its linear dispersion, but also because of the still rather unclear mechanism of its occurrence in the AC${}_{x}$. The mechanism of formation of the AP on a metal surface was resolved a long time ago [41,42,43]. In that case, a partially filled surface state supports a 2D plasmon, which hybridizes with the ordinary surface plasmon, resulting in the transformation of the usual 2D plasmon with square-root 2D dispersion into a linear AP [41]. However, our previous studies of plasmons in AC${}_{x}$ systems show that the AP is very sensitive and strongly dependent on several parameters: the type of the alkali metal atom (A), the coverage (*x*) and the electrostatic doping. In other words, it is substantially unstable, and the aim of this research is to explore the mechanism of the (in)stability of the AP in the AC${}_{x}$.

In order to do that, we investigate two model systems of doped graphene, KC${}_{8}$ and CsC${}_{8}$, which are isostructural, but the first system does support the AP, while the second one does not. Moreover, the DP in the KC${}_{8}$ is very sharp, while in the CsC${}_{8}$ it is quite broad, even though the occupancies of the graphene π band and the alkali atom σ band in these two systems are almost identical. We shall show that the AP in the AC${}_{x}$ is not, as previously believed, just a consequence of the hybridization between the intraband 2D plasmons in the π and σ bands, but rather a very subtle interplay between these plasmons and the background screening caused by the interlayer interband C(π)→A(σ) transitions. We shall also demonstrate that the electronic screening coming from the high-energy interband transitions significantly reduces the intensity of the AP in both systems.

We study electronic excitations in the AC${}_{x}$ using two complementary methods: the ab initio random phase approximation (RPA) approach [24] and a reduced model. In the reduced model, the electrons in the alkali atom layer are approximated by a parabolic band with parameters (the effective mass $m_{\sigma }$ and the Fermi energy $E_{F\sigma }$) taken from the ab initio calculations. We refer to this approximation as the “massive” 2D electron gas (m2DEG). The graphene band structure is approximated by the conical π bands with the occupation corresponding to the occupation of the π band (Fermi energy $E_{F\pi }$) in the ab initio AC${}_{8}$ sample. This approximation is also known as the ’massless’ Dirac fermion (MDF) approximation [44,45]. The polarization effects coming from the high-energy interband transitions are included through the polarizability parameters $\alpha_{m}$ and $\alpha_{g}$ deduced from the ab initio calculations.

The paper is organized as follows. In Section 2, we present the geometry of the system, a derivation of the ab initio ground state and the RPA spectra of the electronic excitations S(Q,ω) in the AC${}_{x}$ deposited on a dielectric substrate. We also present a theoretical formulation of the reduced model. In Section 3, we show the band structures of the KC${}_{8}$ and CsC${}_{8}$, as well as the intensities of the electronic excitations in these systems. After that, using a detailed analysis, which combines the ab initio RPA method and the reduced model, we resolve the mechanism of the AP instability. The conclusions are presented in Section 4.

## 2. Theoretical Formulation

In this section, we derive the spectral function of the electronic excitations in the self-standing KC${}_{8}$ and CsC${}_{8}$. However, to demonstrate the robustness of these electronic excitations, we show how a dielectric substrate can be included in our expressions for the spectral function, to enable the calculation of the electronic excitation spectra for systems consisting of crystals (CsC${}_{8}$ and KC${}_{8}$) deposited on Al${}_{2}$O${}_{3}$ substrate described by a local dielectric function $\epsilon_{S}\left(\omega \right)$. Additionally, we make use of the plausible assumption [25] that the Al${}_{2}$O${}_{3}$ substrate does not affect the ground state electronic structure (Kohn–Sham (KS) wave functions and energies) of the AC${}_{8}$ crystal, but it can influence its dielectric properties.

Figure 1 shows the geometry of the studied system. The coordinates are oriented so that the AC${}_{8}$ crystal is positioned in the x−y plane, the *z* direction is perpendicular to the crystal plane, the graphene layer occupies the z=0 plane, the alkali atom layer occupies the z=−d plane and the dielectric substrate occupies the z<−h half-space.

### 2.1. Calculation of the Surface Electronic Excitations Spectra

We shall briefly describe the method of calculation of the surface electronic excitation spectra at the arbitrary position z>0, previously used in several studies of electronic excitations in 2D crystals [[22],[23],,[46],[47],[48],[49],[50]].

We start from the 3D Fourier transform of the noninteracting (free) electron response function
(1)$$\begin{array}{c} \chi_{GG^{\prime }}^{0}\left(Q,\omega \right)=\frac{2}{\Omega }\sum_{K\in SBZ}\sum_{n,m}\;\frac{f_{n}\left(K\right)-f_{m}\left(K+Q\right)}{\omega +i\eta +E_{n}\left(K\right)-E_{m}\left(K+Q\right)}\\ \times \rho_{nK,mK+Q}\left(G\right)\;\rho_{nK,mK+Q}^{*}\left(G^{\prime }\right), \end{array}$$
where Ω=S×L is the normalization volume and $f_{nK}={\left[e^{(E_{nK}-E_{F})/kT}+1\right]}^{-1}$ is the Fermi–Dirac distribution at temperature *T*. The matrix elements (or the charge vertices) are
(2)$$\rho_{nK,mK+Q}\left(G\right)={\langle \Phi_{nK}\left|e^{-i(Q+G)\cdot r}\right|\Phi_{nK+Q}\rangle }_{V},$$
where $K=(K_{x},K_{y})$ is the 2D wave vector, Q is the momentum transfer vector parallel to the x−y plane, $G=(G_{\|},G_{z})$ are the 3D reciprocal lattice vectors and r=(ρ,z) is the 3D position vector. The integration is performed over the normalization volume Ω. The plane wave expansion of the wave function has the form
$$\Phi_{nK}\left(\rho ,z\right)=\frac{1}{\sqrt{\Omega }}e^{iK\cdot \rho }\;\sum_{G}C_{nK}\left(G\right)e^{iG\cdot r},$$
where the coefficients $C_{nK}$ are obtained by solving the KS equations within the local density approximation (LDA) self-consistently.

The Fourier expansion of the free electron response function in the *z* and $z^{\prime }$ coordinates is:(3)$$\chi_{G_{\|}G_{\|}^{\prime }}^{0}\left(Q,\omega ,z,z^{\prime }\right)=\frac{1}{L}\sum_{G_{z}G_{z}^{\prime }}\chi_{GG^{\prime }}^{0}\left(Q,\omega \right)e^{iG_{z}z-iG_{z}^{\prime }z^{\prime }},$$
where we assume that our system is periodical in the *z* and $z^{\prime }$ direction as well, i.e., that it repeats periodically from supercell to supercell, and the supercells are q2D crystals separated by the distance *L*. We now need to determine the screened response function $\chi_{G_{\|}G_{\|}^{\prime }}^{0}\left(Q,\omega ,z,z^{\prime }\right)$ of one supercell without including the polarization of the surrounding supercells. This spurious interaction with the replicas of the q2D crystal can be eliminated easily, by using the RPA Dyson equation
(4)$$\begin{array}{c} \chi_{G_{\|}G_{\|}^{\prime }}\left(Q,\omega ,z,z^{\prime }\right)=\chi_{G_{\|}G_{\|}^{\prime }}^{0}\left(Q,\omega ,z,z^{\prime }\right)+\sum_{G_{\|1}G_{\|2}}\int_{-L/2}^{L/2}dz_{1}dz_{2}\;\\ \;\times \chi_{G_{\|}G_{\|1}}^{0}\left(Q,\omega ,z,z_{1}\right)\;v_{G_{\|1}G_{\|2}}^{2D}\left(Q,z_{1},z_{2}\right)\chi_{G_{\|2},G_{\|}^{\prime }}\left(Q,\omega ,z_{2},z^{\prime }\right), \end{array}$$
where the matrix of the bare Coulomb interaction is
(5)$$v_{G_{\|1}G_{\|2}}^{2D}\left(Q,z,z^{\prime }\right)=v_{G_{\|1}}^{2D}\left(Q,z,z^{\prime }\right)\delta_{G_{\|1}G_{\|2}},$$
and the 2D Fourier transform of the bare Coulomb interaction is
(6)$$v_{G_{\|1}}^{2D}\left(Q,z,z^{\prime }\right)=\frac{2\pi }{|Q+G_{\|1}|}e^{-|Q+G_{\|1}||z-z^{\prime }|}.$$

Since the integrations in (Equation 4) are performed from −L/2 to L/2, the interaction between the density fluctuations, via the Coulomb propagator $v_{G_{\|1}G_{\|2}}^{2D}\left(Q,z,z^{\prime }\right)$, is limited to one supercell located at −L/2<z<L/2. After inserting the Fourier expansion (Equation 3), and a similar one for χ, in RPA Dyson Equation (Equation 4), it again becomes a matrix equation
(7)$$\chi_{GG^{\prime }}\left(Q,\omega \right)=\chi_{GG^{\prime }}^{0}\left(Q,\omega \right)+\sum_{G_{1}G_{2}}\chi_{GG_{1}}^{0}\left(Q,\omega \right)\;v_{G_{1}G_{2}}^{2D}\left(Q\right)\chi_{G_{2}G^{\prime }}\left(Q,\omega \right),$$
where the matrix of the bare Coulomb interaction is
(8)$$\begin{array}{c} v_{G_{1}G_{2}}^{2D}\left(Q\right)=v_{G_{1}G_{2}}^{3D}\left(Q\right)-p_{G_{z1}}p_{G_{z2}}\frac{4\pi (1-e^{-\left|Q+G_{\|1}\right|L})}{\left|Q+G_{\|1}\right|L}\\ \times \frac{{\left|Q+G_{\|1}\right|}^{2}-G_{z1}G_{z2}}{\left({\left|Q+G_{\|1}\right|}^{2}+G_{z1}^{2}\right)\left({\left|Q+G_{\|1}\right|}^{2}+G_{z2}^{2}\right)}\delta_{G_{\|1}G_{\|2}}, \end{array}$$
with $v_{G_{1}G_{2}}^{3D}\left(Q\right)=\frac{4\pi }{{\left|Q+G_{1}\right|}^{2}}\delta_{G_{1}G_{2}}$, $G_{z}=\frac{2\pi k}{L}$, $p_{G_{z}}={\left(-1\right)}^{k}$, and k∈Z.

The solution of Equation (Equation 7) has the form
(9)$$\chi_{GG^{\prime }}\left(Q,\omega \right)=\sum_{G_{1}}E_{GG_{1}}^{-1}\left(Q,\omega \right)\chi_{G_{1}G^{\prime }}^{0}\left(Q,\omega \right),$$
where the dielectric matrix is
(10)$$E_{GG^{\prime }}\left(Q,\omega \right)=\delta_{GG^{\prime }}-\sum_{G_{1}}V_{GG_{1}}^{2D}\left(Q\right)\chi_{G_{1}G^{\prime }}^{0}\left(Q,\omega \right).$$

After solving Equation (Equation 7), the nonlocal screened response function in the *z* and $z^{\prime }$ direction becomes:(11)$$\chi_{G_{\|}G_{\|}^{\prime }}\left(Q,\omega ,z,z^{\prime }\right)=\frac{1}{L}\sum_{G_{z}G_{z}^{\prime }}\chi_{GG^{\prime }}\left(Q,\omega \right)e^{iG_{z}z-iG_{z}^{\prime }z^{\prime }}.$$

The propagator of the induced dynamically screened Coulomb interaction can be calculated from the response function (Equation 9) as [50]
(12)$$W_{G_{\|}}^{ind}\left(Q,\omega ,z,z^{\prime }\right)=\int_{-L/2}^{L/2}dz_{1}dz_{2}v_{G_{\|}}^{2D}\left(Q,z,z_{1}\right)\chi_{G_{\|}0}\left(Q,\omega ,z_{1},z_{2}\right)\;v_{0}^{2D}\left(Q,z_{2},z^{\prime }\right),$$
where the index zero means that $G_{\|}^{\prime }=0$. After using the expansion (Equation 11) and Equation (Equation 6), the integrations over $z_{1}$ and $z_{2}$ can be performed analytically, and the induced dynamically screened interaction at $z,z^{\prime }>0$ can be written as
(13)$$W_{G_{\|}}^{ind}\left(Q,\omega ,z,z^{\prime }\right)=e^{-|Q+G_{\|}|z-Qz^{\prime }}D_{G_{\|}}\left(Q,\omega \right)$$
where the propagator of the surface excitations is
(14)$$D_{G_{\|}}\left(Q,\omega \right)=\sum_{G_{z1}G_{z2}}\chi_{G_{\|},0,G_{z1},G_{z2}}\left(Q,\omega \right)F_{G_{z1}}\left(Q+G_{\|}\right)F_{G_{z2}}^{*}\left(Q\right),$$
and the form factors *F* are
(15)$$F_{G_{z}}\left(Q\right)=\frac{4\pi p_{G_{z}}}{Q\sqrt{L}}\;\frac{\sinh(\frac{QL}{2})}{Q+iG_{z}}.$$

The spectral function, which defines the intensity of the energy loss by an external perturbation to the excitation of the (Q,ω) modes, can now be calculated as
(16)$$S\left(Q,\omega \right)=-ImD_{G_{\|}=0}\left(Q,\omega \right).$$

Up to this point, we derived the expressions for the self-standing q2D systems, but including a dielectric substrate polarization is now straightforward. It is obtained simply by replacing the bare Coulomb interaction (Equation 6) with the Coulomb interaction screened by the substrate
(17)$$v_{G_{\|}}^{2D}\left(Q,z,z^{\prime }\right)\;\to \;{\tilde{v}}_{G_{\|}}^{2D}\left(Q,\omega ,z,z^{\prime }\right)=v_{G_{\|}}^{2D}\left(Q,z,z^{\prime }\right)+\frac{2\pi }{|Q+G_{\|}|}D_{S}\left(\omega \right)e^{-|Q+G_{\|}|\left(2h+z+z^{\prime }\right)},$$
where the dielectric surface response function is
$$D_{S}\left(\omega \right)=\frac{1-\epsilon_{S}\left(\omega \right)}{1+\epsilon_{S}\left(\omega \right)}.$$

This also means that the matrix (Equation 8) used in matrix Dyson Equation (Equation 7) should be modified as
(18)$$\begin{array}{c} v_{G_{1}G_{2}}^{2D}\left(Q\right)\;\to \;{\tilde{v}}_{G_{1}G_{2}}^{2D}\left(Q,\omega \right)=\;\\ v_{G_{1}G_{2}}^{2D}\left(Q\right)+\frac{|Q+G_{\|1}|}{2\pi }e^{-2|Q+G_{\|1}|h}D_{S}\left(\omega \right)\;F_{G_{z1}}(|Q+G_{\|1}\left|\right)F_{G_{z2}}^{*}(|Q+G_{\|1}\left|\right)\delta_{G_{\|1}G_{\|2}}. \end{array}$$

### 2.2. Calculation of the 2D Dynamical Polarizability Function α(ω)

In the long-wavelength or optical limit (Q→0), the in-plane dielectric function of a 2D crystal can be approximated as
(19)ϵ(Q,ω)=1+2πQα(ω).

The 2D polarizability α can be divided into intraband and interband contributions
(20)$$\alpha \left(\omega \right)=\alpha^{\mathrm{intra}}\left(\omega \right)+\alpha^{\mathrm{inter}}\left(\omega \right),$$
where
(21)$$\alpha^{\mathrm{intra}/\mathrm{inter}}\left(\omega \right)=i\sigma_{\mu }^{\mathrm{intra}/\mathrm{inter}}\left(\omega \right)/\omega ,$$
and μ=x or *y*. The intraband optical conductivity is [51]
(22)$$\sigma_{\mu }^{\mathrm{intra}}\left(\omega \right)=i\frac{e^{2}}{m}\;\frac{n_{\mu }}{\omega +i\eta_{intra}},$$
where the effective number of charge carriers is
(23)$$n_{\mu }\;=\;-\frac{m}{Se^{2}}\;\sum_{n}\sum_{K\in 1.SBZ}\;\frac{\partial f_{nK}}{\partial E_{nK}}\;{\left|j_{nK,nK}^{\mu }\right|}^{2}.$$

The interband optical conductivity is determined from the optical limit of the nonlocal interband conductivity
$$\sigma_{\mu }^{inter}\left(\omega \right)=L\sigma_{\mu }^{inter}\left(\omega ,Q\to 0\right).$$

The nonlocal interband conductivity is [51]
(24)$$\sigma_{\mu }^{\mathrm{inter}}\left(Q,\omega \right)\;=\;-\frac{i\hbar }{\Omega }\sum_{n\neq m}\sum_{K\in 1.SBZ}\frac{{\left|j_{nK,mK+Q}^{\mu }\right|}^{2}}{E_{nK}-E_{mK+Q}}\frac{f_{nK}-f_{mK+Q}}{\hbar \omega +i\eta_{inter}+E_{nK}-E_{mK+Q}},\;$$
where the current vertices are
(25)$$j_{nK,mK+Q}^{\mu }\;=\;\int_{\Omega }dre^{-iQ\cdot r}\;j_{nK,mK+Q}^{\mu }\left(r\right),\;$$
and the current produced by the transitions between the Bloch states $\phi_{nK}^{*}\to \phi_{mK+Q}$ is defined as
$$j_{nK,mK+Q}^{\mu }\left(r\right)\;=\frac{e\hbar }{2im}\left\{\phi_{nK}^{*}\left(r\right)\partial_{\mu }\phi_{mK+Q}\left(r\right)-\left[\partial_{\mu }\phi_{nK}^{*}\left(r\right)\right]\phi_{mK+Q}\left(r\right)\right\}.\;$$

Figure 2 shows the interband contribution to the dynamical polarizability $\alpha^{\mathrm{inter}}\left(\omega \right)$ in the KC${}_{8}$ (black), CsC${}_{8}$ (orange) and doped graphene (red dashed). The graphene is doped so that the Fermi level is 1eV above the Dirac point, which corresponds to the doping of the π bands in the KC${}_{8}$ and CsC${}_{8}$. All three systems show qualitatively equal behavior; the peak at about ω=2 eV, indicating the onset for the $\pi \to \pi^{*}$ interband transitions, and the dip at ω≈4 eV is a consequence of the high density of the $\pi \to \pi^{*}$ interband transitions at the M point of the Brillouin zone. Even though the π bands in all three systems are almost equally doped, we can see a substantial difference in the KC${}_{8}$ and CsC${}_{8}$ statical polarizabilities which are $\alpha^{\mathrm{inter}}\left(\omega =0\right)=2.2$ Å and $\alpha^{\mathrm{inter}}\left(\omega =0\right)=3.05$ Å, respectively. This difference probably comes from the difference in the intensities of the C(π)→A(σ) interlayer (interband) excitations in the two systems. These excitations are manifested as the peak at ω≈0.25 eV for the CsC${}_{8}$, which does not exist for the KC${}_{8}$. Finally, we can see that the agreement between the dynamical polarizabilities in the KC${}_{8}$ and the equivalently doped graphene is almost perfect. As we shall demonstrate later, this small deviation in the low-energy part of the dynamical polarizability is responsible for the disappearance of the AP in the CsC${}_{8}$, but only if we take into account that these transitions represent a perpendicular polarization.

### 2.3. Calculation of the Substrate Dielectric Function

We assume that the dielectric media is vacuum (i.e., $\epsilon_{0}=1$), and that the substrate is aluminium oxide Al${}_{2}$O${}_{3}$ described by the macroscopic dielectric function $\epsilon_{s}\left(\omega \right)$. To calculate the $\epsilon_{s}\left(\omega \right)$, we start from the 3D Fourier transform of the independent electron response function
(26)$$\chi_{GG^{\prime }}^{0}\left(q,\omega \right)=\frac{2}{\Omega }\sum_{k\in 1.\mathrm{BZ}}\sum_{n,m}\;\frac{f_{n}\left(k\right)-f_{m}\left(k+q\right)}{\omega +i\eta +E_{n}\left(k\right)-E_{m}\left(k+q\right)}\rho_{nk,mk+q}\left(G\right)\;\rho_{nk,mk+q}^{*}\left(G^{\prime }\right),$$
where k∈1.BZ indicates that the summation is performed within the first Brillouin zone. The charge vertices are defined as
(27)$$\rho_{nk,mk+q}\left(G\right)=\int_{\Omega }dr\;\phi_{nk}^{*}\left(r\right)e^{-i(q+G)\cdot r}\phi_{mk+q}\left(r\right),$$
where $k=(k_{x},k_{y},k_{z})$, $q=(q_{x},q_{y},q_{z})$ and $G=(G_{x},G_{y},G_{z})$ are the 3D wave vector, the transfer wave vector and the reciprocal lattice vector, respectively. The integration is performed over the normalization volume Ω. We use the response matrix (Equation 26) to determine the dielectric matrix as
(28)$$E_{GG^{\prime }}\left(q,\omega \right)=\delta_{GG^{\prime }}-\sum_{G_{1}}v_{GG_{1}}\left(q\right)\chi_{G_{1}G^{\prime }}^{0}\left(q,\omega \right),$$
where the bare Coulomb interaction is $v_{GG^{\prime }}\left(q\right)=\frac{4\pi }{{\left|q+G\right|}^{2}}\;\delta_{GG^{\prime }}$. Finally, the macroscopic dielectric function is determined by inverting the dielectric matrix
(29)$$\epsilon_{s}\left(\omega \right)=\epsilon_{1}\left(\omega \right)+i\epsilon_{2}\left(\omega \right)=1/E_{G=0G^{\prime }=0}^{-1}\left(q\approx 0,\omega \right).$$

### 2.4. Reduced Model

Analytical modeling of the energy loss spectra is achieved by representing each of the systems, CsC${}_{8}$ and KC${}_{8}$, by a two-layer structure consisting of a single sheet of doped graphene and an m2DEG, placed in vacuum at a distance *d* apart, as shown in Figure 3. In the reduced model, the substrate is neglected, i.e., we assume $\epsilon_{S}\left(\omega \right)=1$, and the energy loss function −ℑ{1/ϵ(Q,ω)} is then obtained from the effective 2D dielectric permittivity for this two-layer structure, in the RPA given by [44,52]
(30)$$\epsilon \left(Q,\omega \right)=\frac{1}{2}\left[1+\coth\left(Qd\right)-2v_{Q}\chi_{g}^{0}\right]-\frac{1}{2}\frac{\;{\mathrm{cosech}}^{2}\left(Qd\right)}{1+\coth\left(Qd\right)-2v_{Q}\chi_{m}^{0}},$$
where $v_{Q}=2\pi /Q$ is the Coulomb interaction, while $\chi_{g}^{0}\left(Q,\omega \right)$ and $\chi_{m}^{0}\left(Q,\omega \right)$ are the response functions of the noninteracting electrons in the graphene and the m2DEG layers, respectively.

For the doped graphene, we follow the method proposed by Gjerding et al. [53], and write the response function as $\chi_{g}^{0}\left(Q,\omega \right)=\chi_{\mathrm{Dirac}}\left(Q,\omega \right)-\alpha_{g}Q^{2}$, where $\chi_{\mathrm{Dirac}}$ is the response function given in Refs. [54,55], which describes both intraband and low-energy interband electron transitions within the π electron bands approximated by the Dirac cones with the Fermi energy $E_{F\pi }$, while $\alpha_{g}$ is the phenomenological parameter providing the correction due to the high-energy interband transitions. For the m2DEG, we similarly write $\chi_{m}^{0}\left(Q,\omega \right)=\chi_{2}\mathrm{DEG}\left(Q,\omega \right)-\alpha_{m}Q^{2}$, where $\chi_{2}\mathrm{DEG}$ is the polarization function given in Ref. [56], describing the intraband transitions in the 2DEG which occupies a single parabolic energy band with the effective mass $m_{\sigma }$ and the Fermi energy $E_{F\sigma }$, while $\alpha_{m}$ is a phenomenological parameter taking into account interband transitions in the m2DEG. The expressions used for both response functions, $\chi_{g}^{0}\left(Q,\omega \right)$ and $\chi_{m}^{0}\left(Q,\omega \right)$, are formulated for zero temperature, but they are corrected by the Mermin procedure to take into account a finite damping parameter η in both graphene and m2DEG layers [53]. We use η=40 meV as in the ab initio calculations, as well as η=65 meV to take into account the additional smearing at room temperature.

We note that the relevant parameters for the KC${}_{8}$ (d=2.92 Å, $E_{F\pi }=1.01$ eV, $E_{F\sigma }=0.9$ eV, $m_{\sigma }=0.92$) and CsC${}_{8}$ (d=3.13 Å, $E_{F\pi }=1.03$ eV, $E_{F\sigma }=1.03$ eV, $m_{\sigma }=0.72$) are obtained from the electronic band structure calculation for these two systems, shown in Figure 4a,b, respectively. The parameters α are obtained from the static limit (ω→0) of the ab initio results for the corresponding dynamic polarizability functions $\alpha^{\mathrm{inter}}\left(\omega \right)$ in the optical limit. In particular, the value $\alpha_{g}\approx 1.3$ Å for high-energy interband transitions is deduced from the data for the intrinsic graphene [53], and by adding the value for the low-energy $\pi \to \pi^{*}$ interband transitions, estimated from the Dirac model in the optical limit at zero temperature [57] as ${\left(4\pi E_{Fg}\right)}^{-1}\approx 1.15$ Å, we obtain that the total contribution of the interband transitions in the doped graphene is around 2.45 Å. This is close to the result $\alpha^{\mathrm{inter}}\left(\omega =0\right)=2.2$ Å shown in Figure 2 for the KC${}_{8}$, which indicates that $\alpha_{m}\approx 0$ for that system. On the other hand, the result $\alpha^{\mathrm{inter}}\left(\omega =0\right)=3.05$ Å shown in Figure 2 for CsC${}_{8}$ indicates that $\alpha_{m}\approx 0.6$ Å for that system.

### 2.5. Computational Details

The KC${}_{8}$, CsC${}_{8}$ and graphene KS wave functions $\phi_{nK}$ and energies $E_{nK}$ are determined using the plane-wave self-consistent field DFT code (PWSCF) within the QUANTUM ESPRESSO (QE) package [58]. The core–electron interaction is approximated by the norm-conserving pseudopotentials [59,60]. For the KC${}_{8}$ and CsC${}_{8}$, the exchange correlation (XC) potential is approximated by the Perdew–Burke–Ernzerhof (PBE) generalized gradient approximation (GGA) functional [61]. The ground state electronic densities are calculated using the 8×8×1 Monkhorst–Pack K-mesh [62] and the plane-wave cut-off energy is 60Ry. For both AC${}_{8}$ crystals, we used the hexagonal Bravais lattice, where a=4.922 Å and the separation between the AC${}_{8}$ layers is L=2.5a. The atomic and the unit cell relaxations were performed until maximum force below 0.001 Ry/a.u. was obtained. After performing the structural optimization, the obtained separations between the K and Cs layers and the graphene layer are d = 2.92 Å and d = 3.13 Å respectively. The graphene XC potential is approximated by the Perdew–Zunger (PZ) LDA [63]. The ground state electronic density is determined using 21×21×1 K-point mesh, and for the plane-wave cut-off energy we choose 50 Ry. For the graphene unit-cell constant we use the experimental value $a_{g}=2.45$ Å [64], while for the superlattice unit-cell constant we take $L=5a_{g}$.

The AC${}_{8}$ response functions (Equation 1) and conductivities (Equation 22)–(Equation 25) are evaluated from the wave functions $\Phi_{nK}\left(r\right)$ and energies $E_{n}\left(K\right)$ calculated for the 201×201×1 Monkhorst–Pack K-point mesh. The band summation (n,m) is performed over 60 bands, the damping parameters are $\eta =\eta_{intra}=\eta_{inter}=10$ meV and the temperature is T=25 meV. Due to the large spatial dispersivity of the dielectric response in the perpendicular (*z*) direction, the crystal local field effects are taken into account only in the *z* direction and neglected in the x−y plane, i.e., we set the $G_{\|}$ to zero. To calculate the matrix $\chi_{G_{z},G_{z}^{\prime }}$ we set the energy cut-off to 10Ry, which corresponds to a 23×23 matrix. Graphene conductivity (Equation 22)–(Equation 25) is calculated by performing the summation over 601×601×1 K-point mesh, the band summations (n,m) are performed over 20 bands, the damping parameters are $\eta_{\mathrm{intra}}=10$ meV and $\eta_{\mathrm{inter}}=25$ meV and the temperature is T=25 meV. The conductivity of the doped graphene is calculated using the rigid band approximation, i.e., the occupation parameter (the Fermi energy relative to the Dirac point) is adjusted a posteriori.

The ground state electronic density of the bulk Al${}_{2}$O${}_{3}$ is calculated using 9×9×3 K-mesh, the plane-wave cut-off energy is 50Ry and the Bravais lattice is hexagonal (12 Al and 18 O atoms in the unit cell) with the lattice constants a=4.76 Å and c=12.99 Å. The response function (Equation 26) of the Al${}_{2}$O${}_{3}$ is calculated using the 21×21×7*k*-point mesh and the band summations (n,m) are performed over 120 bands. The damping parameter is η=100 meV and the temperature is T=10meV. For the optically small wave vectors q≈0, the crystal local field effects are negligible, so the crystal local field effect cut-off energy is set to zero. Using this approach, the Al${}_{2}$O${}_{3}$ dielectric function is estimated to be $\epsilon_{S}\left(\omega \right)\approx \epsilon_{S}\left(\omega =0\right)=2.9$ for ω<2 eV, which is in good agreement with the experimental value 3.4 [65]. However, in the calculations we used the full dynamical $\epsilon_{S}\left(\omega \right)$.

## 3. Results and Discussion

Figure 4 shows (a) KC${}_{8}$ and (b) CsC${}_{8}$ band structures, respectively, along the high symmetry Γ→K→M→Γ directions. The blue circles represent the parabolic fit $E\left(K\right)=E_{F\sigma }+\frac{\hbar^{2}K^{2}}{2m_{\sigma }}$ for the alkali atom σ bands. In both figures, we can see that the graphene π band and the alkali atom σ band are significantly doped by the electrons, so the Fermi level is around 1eV above the Dirac point and the σ band bottom. It is also evident that the σ bands behave almost as an ideal 2D free electron gas. They are parabolic (described by effective masses $m_{\sigma }=0.916$ and $m_{\sigma }=0.72$, respectively) up to the Fermi energy, especially in Figure 4a where the σ band is parabolic almost through the entire Brillouin zone. Finally, the most important feature is the obvious similarity between the two band structures, which both fulfill conditions for the occurrence of the AP (two bands of different effective masses crossing the Fermi energy) [22,41]. However, as we shall see, the spectra of the low-energy electronic excitations in these two systems are quite different. In both systems, we can identify the lowest band that remains above the Fermi lever for all wave-vectors, and we call it the lowest unoccupied band (LUCB), even though for some wave-vectors there is another unoccupied band below that one, since this one turns out to be of particular importance, as explained below.

Figure 5 shows the intensities of the electronic excitations in the (a) KC${}_{8}$ and (b) CsC${}_{8}$ deposited on the dielectric Al${}_{2}$O${}_{3}$ surface. Since the graphene π bands are abundantly doped by the electrons, both systems support a strong DP. However, although the occupancies of both π bands are almost identical ($E_{F\pi }=1.01$ eV for the KC${}_{8}$ and 1.03 eV for the CsC${}_{8}$), the intensities of the DP are quite different. We can see that the DP in the CsC${}_{8}$ is broader and much more efficiently damped by the interband $\pi \to \pi^{*}$ electron hole excitations than the DP in the KC${}_{8}$. Moreover, we can see that the DP in the KC${}_{8}$ is very sharp and it extends deep into the interband $\pi \to \pi^{*}$ continuum while the DP in the CsC${}_{8}$ decays immediately upon entering the interband $\pi \to \pi^{*}$ continuum. The most interesting difference between the two systems is that, even though the occupancies of their σ bands are very similar ($E_{F\sigma }=0.9$ eV in the KC${}_{8}$ and 1.03 eV in the CsC${}_{8}$), the KC${}_{8}$ does support the AP while the CsC${}_{8}$ does not. Moreover, we can see that the π plasmon (denoted by the blue symbol π) in the KC${}_{8}$ is well defined, and it appears to be a very intensive optically active plasmon (its intensity does not vanish for Q→0), while the π plasmon in the CsC${}_{8}$ is less intensive, more diffuse and not in optically active mode. It is very interesting that these two very similar band structures lead to very different excitation spectra.

To demonstrate that these effects are not driven by the dielectric substrate, in Figure 5 we also show the intensities of the electronic excitations in the self-standing (c) KC${}_{8}$ and (d) CsC${}_{8}$. The magenta dots in Figure 5a,b denote the DP and AP dispersion relations in the self-standing samples, for comparison. It is worth noting that the insulator surface does not change the qualitative behavior of the electronic modes. The only quantitative differences are: the dielectric screening slightly reduces the DP energy and slightly reduces the intensities of the AP and π plasmons. However, the AP in the KC${}_{8}$ is still clearly visible. Since the substrate does not affect the qualitative behavior of the excitation spectra, it will be omitted from further consideration. Therefore, the fact that one system does and the other does not support the AP, even though the bands crossing the Fermi energy in the two systems are almost identical, suggests that the mechanism responsible for this AP instability is probably in the screening coming from the interband excitations beyond the Fermi energy.

To investigate this mechanism, we take advantage of the reduced model which clearly distinguishes between the different interband contributions to the dynamical response in the two systems. As discussed in Section 2.4, the statical polarizabilities due to the high-energy interband excitations in the metallic subsystem (characterized by the parameter $\alpha_{m}$) are $\alpha_{m}\approx 0$ in the KC${}_{8}$ and 0.6 in the CsC${}_{8}$. This difference suggests that this polarization mechanism may be responsible for suppressing the AP. Figure 5e,f show the energy loss function −ℑ{1/ϵ(Q,ω)} in the self-standing KC${}_{8}$ and CsC${}_{8}$, respectively, obtained using the reduced model. We can see that the agreement between the ab initio and the reduced model intensities for the KC${}_{8}$ for ω<3 eV and Q<0.14 a.u. is almost perfect. Beyond these values, the plasmons in the reduced model appear at higher energies, which is reasonable since the reduced model neglects the dynamical effects of the high-energy interband transitions. For example, one can notice the absence of the π plasmon in the reduced model. In the case of the CsC${}_{8}$, the agreement is no longer so good, and the most important difference is that the reduced model (at this level of approximation) is obviously not able to reproduce the disappearance of the AP. The stronger metallic interband screening in the CsC${}_{8}$ is still not sufficient to suppress the AP. Moreover, even the implementation of the ab initio dynamic polarizability $\alpha^{\mathrm{inter}}\left(\omega \right)$ in the reduced model (taking into account that the low-energy interband excitations in the $\chi_{\mathrm{Dirac}}$ and $\chi_{2}\mathrm{DEG}$ need to be extracted) fails to cause the disappearance of the AP. This means that the mechanism of the AP disappearance is more complex and obviously the explanation requires more accurate treatment of the spatial dispersivity of the interband dynamical response, beyond the 2D model. Therefore, we shall again exploit the ab initio method, which incorporates the effects of the local crystal field (the spatial dispersivity of the dynamical response in the *z* direction), where the interband screening will be modified by changing the number of valence bands participating in the interband screening.

### Resolving the Mechanism of the AP Instability

To explore how the screening coming from the interband excitations beyond the Fermi energy influences the AP, we omit one or more unoccupied bands from the calculation, to determine exactly how each band influences the excitation spectra. As we can see from the band structures shown in Figure 4a,b, in both systems, CsC${}_{8}$ and KC${}_{8}$, there is one band crossing the Fermi level, i.e., for some wave-vectors that band is the highest occupied valence band (HOVB), while for other wave-vectors that same band is the lowest unoccupied band. Therefore, to avoid confusion, we use the expression lowest unoccupied band for the next one, i.e., the first band above the Fermi level at the Γ point (magenta line in Figure 4a,b), since that one remains above the Fermi level for all wave-vectors. That particular band is the one that has to be omitted from the calculation to achieve a qualitative difference with respect to the complete calculation (the one taking into account all bands). However, the difference becomes much more significant if we omit more bands. To keep track of the bands omitted from the calculation, we introduce the integer *n* which denotes the number of omitted bands. For example, n=0 denotes that no bands are omitted, n=1 denotes that only the lowest unoccupied band is omitted, n=2 denotes that the first two unoccupied bands are omitted, etc.

Figure 6a,b show how the AP peak changes as we omit the unoccupied bands from the calculation, for Q=0.1 a.u. and for the CsC${}_{8}$ and KC${}_{8}$, respectively. The thick black lines show the actual spectra, i.e., the ones obtained without omitting any bands (n=0), and we can see that for the selected wave-vector the AP in the KC${}_{8}$ is clearly present (though not very strong), while in the CsC${}_{8}$ it does not exist. On the other hand, the red line shows that the AP peak exists in both systems if we omit the LUCB from the calculation (n=1). In the CsC${}_{8}$, that peak is very weak but it exists, while in the KC${}_{8}$ it is, surprisingly, lower than the actual (n=0) peak. The other lines show that the intensity of the peak increases as we omit more and more bands, in both systems, and it is the strongest when all the unoccupied bands are omitted (n=∞). The same can be seen in Figure 6c, which shows how the intensity of the peak changes with *n*. It is important to point out that we also attempted to keep the lowest unoccupied band in the calculation while omitting any number of the other unoccupied bands, above that one, and that did not lead to the occurrence of the AP in the CsC${}_{8}$. This means that omitting the lowest unoccupied band (n=1) is crucial for the occurrence of the AP, while omitting the higher bands as well (n>1) only enhances its intensity. This leads to the conclusion that the mechanism responsible for the disappearance of the AP in the CsC${}_{8}$ is the small difference in the ‘out-of-plane’ polarization coming from the interband (interlayer) transitions between the graphene π band and the lowest unoccupied alkali atom σ band (denoted by red arrows in Figure 4). This difference is also manifested as the small peak at ω≈0.25 eV in the CsC${}_{8}$ dynamical polarizability $\alpha^{\mathrm{inter}}\left(\omega \right)$ shown in Figure 2, which is missing for the KC${}_{8}$. However, even if we include this difference in the dynamical polarizability in the reduced model, that still fails to reproduce the disappearance of the AP. This is because the reduced model is inherently a 2D model, i.e., it allows only the ‘in-plane’ polarization, while the π→σ transitions induce the ‘out-of-plane’ polarization.

Another interesting point is that the difference in the intensities of the π→σ transitions is obviously correlated with the strength of the hybridization of the π and σ bands at the crossing point. We can see that the π and σ bands in the KC${}_{8}$ intersect without any distortion while in the CsC${}_{8}$ we notice a significant avoided crossing. In other words, the π and σ in the CsC${}_{8}$ hybridize significantly while this is not the case in the KC${}_{8}$. This leads to one very unusual conclusion; the intensity of the AP, which was to date believed to be a consequence of various long-range screening mechanisms, can be significantly affected by the short-range electronic correlations occurring between the atomic orbitals (i.e., by the chemical bonding between the alkaline atoms and graphene).

## 4. Conclusions

We analyzed the origin of AP instability in graphene doped by alkali metals on two prototype systems, KC${}_{8}$ and CsC${}_{8}$. Even though the band structures of these systems are almost identical, we proved that the hybridization between the C(π) and the K(σ) or Cs(σ) bands at the crossing point causes a significant modification of the dynamic polarizability in the perpendicular direction, which has a substantial influence on the low-energy excitation spectra in these systems. We demonstrated that the net perpendicular screening in the CsC${}_{8}$ causes the disappearance of the AP and significant weakening of the DP. We also demonstrated that the electronic screening coming from the high-energy interband transitions (beyond the HOVB–LUCB interval) significantly reduces the intensity of the AP in both systems, KC${}_{8}$ and CsC${}_{8}$. This illustrates the importance of the nature of the chemical bonding between the alkaline atoms and graphene, as well as the importance of perpendicular dispersivity of the dynamical response in theoretical simulations of technologically interesting low-energy plasmons in chemically doped graphenes. 

## Figures and Tables

**Figure 1 ijms-23-04770-f001:**
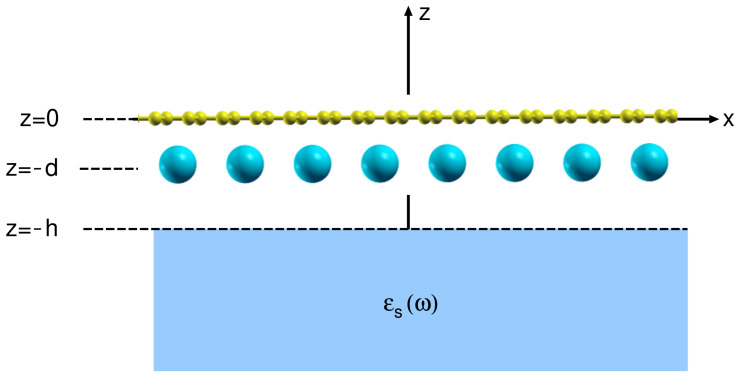
Schematic representation of the AC${}_{8}$ crystal deposited on a dielectric substrate described by a local dielectric function $\epsilon_{S}\left(\omega \right)$. The substrate (blue) occupies the region z<−h, the graphene layer (yellow) is in the z=0 plane and the alkali atom (turquoise) layer is in the z=−d plane.

**Figure 2 ijms-23-04770-f002:**
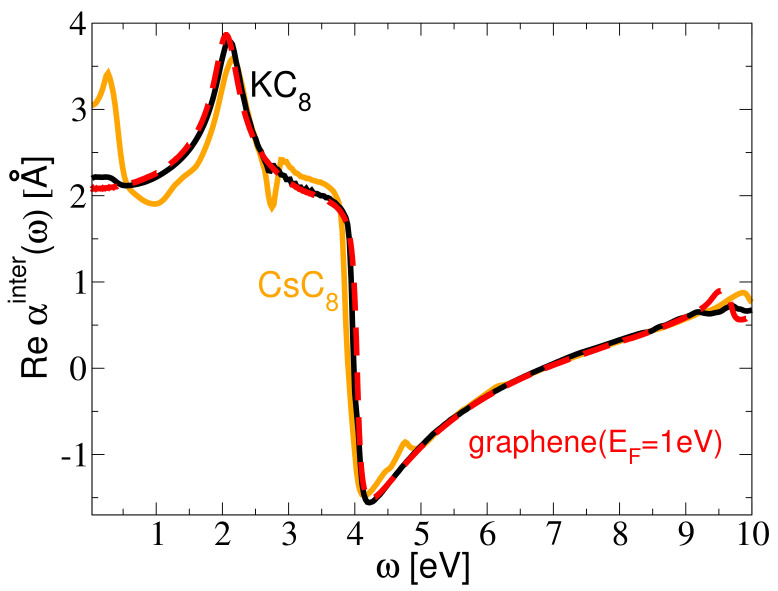
The interband contribution to the dynamical polarizability $\alpha^{\mathrm{inter}}\left(\omega \right)$ for the KC${}_{8}$ (black), CsC${}_{8}$ (orange) and doped graphene (red dashed). The graphene is doped so that the Fermi energy is 1eV above Dirac point, which corresponds to the doping of the π bands in the KC${}_{8}$ and CsC${}_{8}$.

**Figure 3 ijms-23-04770-f003:**
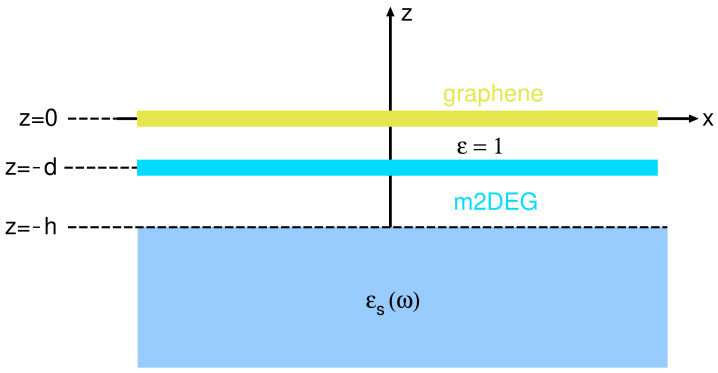
Schematic representation of the reduced model. The alkali atom layer is approximated by ‘massive’ 2D electron gas (parabolic σ band), and the graphene layer is described by the ‘massless’ Dirac fermion (MDF) approximation (conical π band).

**Figure 4 ijms-23-04770-f004:**
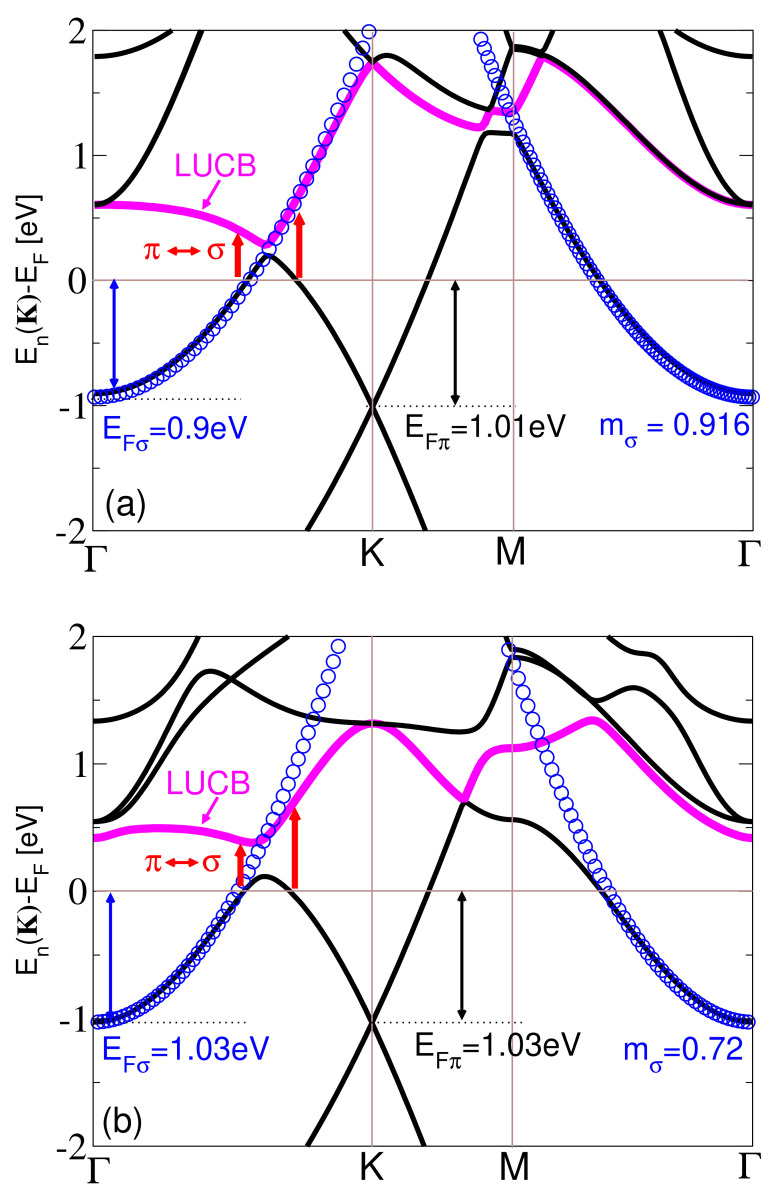
The electronic band structure in (**a**) KC${}_{8}$ and in (**b**) CsC${}_{8}$. The magenta line denotes the lowest unoccupied band (LUCB). The blue circles represent the parabolic fit $E\left(K\right)=E_{F\sigma }+\frac{\hbar^{2}K^{2}}{2m_{\sigma }}$ of the alkali atom σ band.

**Figure 5 ijms-23-04770-f005:**
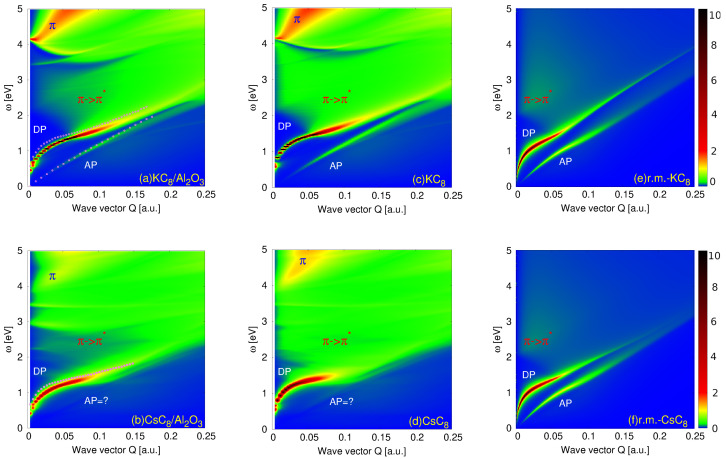
The ab initio spectra of the electronic excitations S(Q,ω) in the (**a**) KC${}_{8}$ and (**b**) CsC${}_{8}$ deposited on dielectric Al${}_{2}$O${}_{3}$ surface with h=5.92 Å and h=6.13 Å respectively (i.e., the separation between the dielectric surface and the alkali atom layer is chosen to be 3 Å). The ab initio spectra of the electronic excitations in the self-standing (**c**) KC${}_{8}$ and (**d**) CsC${}_{8}$. The magenta dots in (**a**,**b**) denote the DP and AP dispersion relations in the self-standing samples, i.e., the DP and AP in (**c**,**d**). The energy loss function −ℑ{1/ϵ(Q,ω)} in the self-standing (**e**) KC${}_{8}$ and (**f**) CsC${}_{8}$ was obtained using the reduced model.

**Figure 6 ijms-23-04770-f006:**
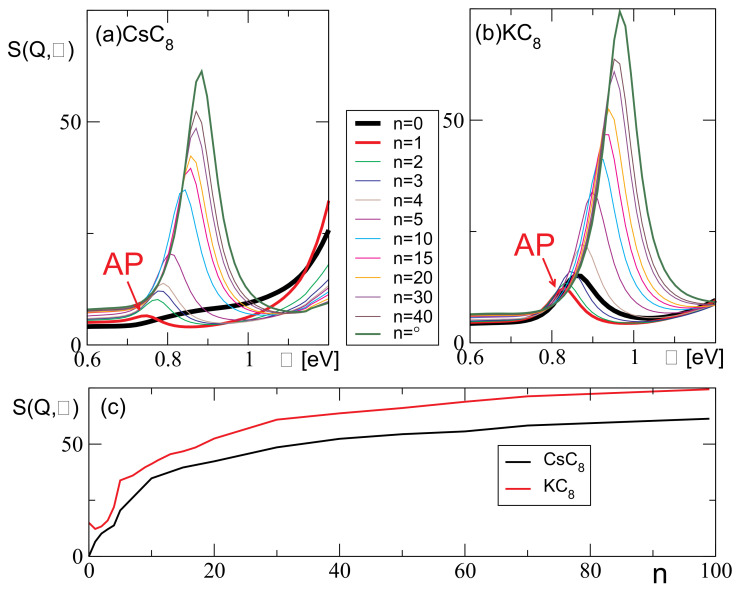
AP peak for Q=0.1a.u. when various numbers of the unoccupied bands are omitted from the calculation (*n*) for the (**a**) CsC${}_{8}$ and (**b**) KC${}_{8}$. (**c**) Intensity of the AP peak as a function of *n* for the CsC${}_{8}$ (black line) and KC${}_{8}$ (red line).

## Data Availability

Not Applicable.

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
