# Peer review of "Resolving the Mechanism of Acoustic Plasmon Instability in Graphene Doped by Alkali Metals"

_ijms, 2022, doi:10.3390/ijms23094770_

Round 1
Reviewer 1 Report
The paper Resolving the Mechanism of the Acoustic Plasmon Instability in the Alkali Metals Doped Graphene prepared by Leonardo Marušić et al., presents new and interesting results that deserve to be published after some minor improvements:
- Please update the introduction and reference list by adding at least 10 references from 2020-2021 in order to highlight better the importance of your work in the field.
- Please use the journal's template. Format the references accordingly.
- Congrats on this interesting work.
Author Response
We want to thank the referee for the fair review of the paper, and for the congratulations. We have corrected the manuscript in accordance to the referee’s recommendations as follows:
- Please update the introduction and reference list by adding at least 10 references from 2020-2021 in order to highlight better the importance of your work in the field.
We thank the referee for this important comment. The introduction was, in fact, a bit out of date, especially the references, so we updated it by adding an additional paragraph (now the first paragraph) and 14 recent references (the first 14 on the new list of references).
- Please use the journal's template. Format the references accordingly.
We have formatted the references using the template required by the journal.
Reviewer 2 Report
Thank you for the good work.
Just few comments:
1) Page 8, Fig. 5 False ruler on the right side of figures, scale numbers are too small. And, panels have No numbering which is mentioned in the Fig. 5 citation.
2) In reference: ref. 13 and ref. 24 is the repetition.
Author Response
We want to thank the referee for the fair review of the paper. We have improved the English and corrected the manuscript in accordance to the referee’s recommendations as follows:
- Page 8, Fig. 5 False ruler on the right side of figures, scale numbers are too small. And, Panels have No numbering which is mentioned in the Fig. 5 citation.
We thank the referee for bringing this to our attention. We have increased the size of the letters. The panel numbering is placed in the lower right corner of each panel.
- In reference: ref. 13 and ref. 24 is the repetition.
We deleted the ref. 24 and updated the referencing accordingly.
Reviewer 3 Report
In this manuscript the authors report an analysis on the origin of the acoustic plasmon instability in the alkali metals doped graphene on two prototype systems, KC8 and CsC8. Present research team has a long experience on these type of studies (see reference 7-12, 30, 32-38). The subject is interesting, the employed methodologies are competently used. Results are well presented and discussed, the conclusions fully consistent with the obtained data and bibliography section is accurate. I suggest to publish in the present form.
Author Response
We want to thank the referee for the fair review of the paper. We have improved the English.